# Ecological change and conflict reduction led to a social circulatory system in ants
Marie-Pierre Meurville[1], Daniele Silvestro[2,3,5] & Adria C. LeBoeuf [1,4,5] ✉

Behavioral innovations can be ecologically transformative for lineages that perform them and for their associated communities. Many ecologically dominant, superorganismal, and speciose ant lineages use mouth-to-mouth social regurgitation behavior – stomodeal trophallaxis – to share exogenous and endogenous materials within colonies. This behavior is less common in other species-poor, less cooperative ant lineages. How and why trophallaxis evolved and fixed in only some ant clades remains unclear, and whether this trait could be indicative of superorganismality has yet to be established. Here we show that trophallaxis evolved in two main events, in non-doryline formicoids around 130 Ma and in some ponerines around 90 Ma, lineages that today encompass 86% of all ant species. We found that trophallaxis evolved in lineages that began drinking sugary liquids and that had reduced intra-colonial conflict by constraining worker reproductive potential. Evolution of trophallaxis increased net diversification. Causal models indicate that trophallaxis required low reproductive conflict and contributed to the large colony sizes of the ants that use it. This suggests that the evolution of social regurgitation was enabled by both social conflict reduction and opportunistic inclusion of nectar and honeydew in the ant diet during the shifts in terrestrial ecosystems toward flowering plants.

Novel behaviors can impact not only a species' own evolutionary trajectory through selection, speciation, and extinction dynamics, but also the evolutionary trajectories of associated species, and even entire ecosystems[1–3]. Occasionally new behaviors and their associated morphology can be so crucial that entire clades exclusively display central adaptations around the behavior[4–7].

Behavioral traits and associated phenotypic adaptations were central to the evolution of one of the most ecologically dominant invertebrate clades: the ants[3,8,9]. The ant clade harbors highly diverse ecologies, morphologies, life-history traits, and behaviors. Some ant species display such high levels of cooperation that they are considered to have passed a major evolutionary transition to superorganismality, characterized by firm reproductive division of labor, low conflict, and high cooperation[10–12]. However, other extant ant species have significant within-colony conflict, where workers can be mated, lay diploid eggs that can develop into new queens, and fundamentally act as the colony's reproductive[13]. Such workers display what we term worker reproductive autonomy, a condition that would threaten colony cohesion, and they do not satisfy Wheeler's conditions for a superorganism[14] (worker lifetime unmatedness). The current dominant view in ant evolution is that the reproductive division of labor was already complete in the common ancestor to all ants[15–17] and any species with totipotent workers are secondary reductions. However, the analyses these conclusions were based on relied on strong unvalidated assumptions and at a time when the ant phylogeny was less resolved (a major subfamily with worker reproductive autonomy, Ectatomminae, was misplaced). Further, once systems undergo major evolutionary transitions in individuality[18], various ratcheting processes typically block them from reverting[19]. Together, these features collectively suggest we should consider the possibility that not all ants are superorganisms, and instead, superorganismality may have evolved one or more times within the ants.

Many ants from diverse subfamilies perform a social regurgitation behavior between adults called stomodeal trophallaxis (hereafter referred to as trophallaxis), wherein workers regurgitate the contents of their crop to nestmates in a mouth-to-mouth interaction[20]. Trophallaxis forms a physiological channel between bodies to transfer liquid material[4]. The transferred social fluid consists of food, but also endogenously produced metabolites and proteins[20–23]. Such body-to-body resource sharing is costly both in terms of resources and immune risks, suggesting that such sharing should predominantly occur between individuals who cooperate rather than compete[4,24]. While resource sharing has not been a frequent component of

[1]Department of Biology, University of Fribourg, Fribourg, Switzerland. [2]Department of Biological and Environmental Sciences, Gothenburg Global Biodiversity Centre, University of Gothenburg, Gothenburg, Sweden. [3]Department of Biosystems Science and Engineering, ETH Zurich, Klingelbergstrasse 48, 4056 Basel, Switzerland. [4]Department of Zoology, University of Cambridge, Cambridge, United Kingdom. [5]These authors contributed equally: Daniele Silvestro, Adria C. LeBoeuf. ✉e-mail: acl79@cam.ac.uk

theory on major evolutionary transitions, in practice, the internal units of collectively developing organisms typically share resources via some form of corporal or extra-corporeal vasculature[25]. Indeed, when resource sharing is implemented in models of the major evolutionary transition to multicellularity, it allows functional specialization to be adaptive even when the fitness returns from division of labor are saturating[26].

Despite trophallaxis being an emblematic ant behavior, the trophallactic behavior of 98.7% of ant species remains unestablished[20], and relatively little has been written on how or why it evolved and what were its evolutionary consequences. We hypothesize that trophallaxis could be a marker of superorganismality in ants, in that it enables a novel form of extra-corporeal vasculature—a social circulatory system. In this study, we tested four scenarios to understand whether trophallaxis behavior predicts or is predicted by traits that approximate ecology, major morphological shifts and superorganismality, and further, we test how this behavior impacted ant diversification rates. We combine deep-learning predictions, phylogenetic comparative methods, and model selection to infer the evolution of trophallaxis behavior across hundreds of ant species from 15 subfamilies and reconstruct the evolution and impact on the long diversification history of this clade.

## Results

To understand the evolution of trophallaxis in concert with superorganismality, social complexity, and ecology, we compiled all records of adult-adult stomodeal trophallaxis from the literature (as of early 2021, 265 papers, 205 species, see database https://osf.io/cuapw/). We also asked experts whether they had observed social regurgitation in species they had worked with but where the behavior had not been reported in literature. Finally, for as many species as possible, while aiming at balancing taxon sampling, we also gathered four additional traits that had been proposed to intertwine with trophallaxis evolution[14,20,27–34], (1) worker reproductive autonomy as a proxy for intracolonial conflict, (2) a diet including sugary liquids, (3) presence of sting, and (4) colony size. While adult-larval trophallaxis[14], cuticle thickness[35], use of trophic eggs[36,37], and frequency of polygyny[38] might also be related traits, too few data were available to perform meaningful tests over a clade of >14,000 species.

### Traits shaping the evolution of trophallaxis

To understand how these four traits correlate with social regurgitation between adults over the evolutionary history of ants, we used a phylogenetic d-test[39] on 212 species where the presence or absence of trophallaxis behavior is known from the literature or from experts. We found that a diet including sugary liquids, lack of worker reproductive autonomy (when workers cannot be functionally mated and produce diploid offspring), lack of sting, and large colony size all correlated with trophallaxis, while small colony size negatively correlated with trophallaxis (Table S1). We found no correlation between trophallaxis and full worker sterility or workers that can only lay male eggs, suggesting that trophallaxis is inhibited only when reproductive conflict impacts colony cohesion—full worker reproductive autonomy would allow straightforward colony fission[40].

To identify causality links in these correlations and better understand the interplay of these traits over evolution, we built four hypothesized scenarios for the evolution of trophallaxis and tested them with phylogenetic path analysis using phylogenetic generalized least squares and d-separation[41] (Fig. 1). In scenario 1 (reduced reproductive conflict, Fig. 1A) the social sharing of metabolism should only occur in species that have sufficiently reduced conflict to cross the major evolutionary transition toward superorganismality[11,42]. If true, trophallaxis should not occur in species with high reproductive conflict, such as that found in species with worker reproductive autonomy (e.g., where workers can mate and produce diploid offspring, as in *Harpegnathos*[43]). This hypothesis predicts a strong negative correlation between worker reproductive autonomy and the use of trophallaxis, which could come about differently depending on the ancestral worker's reproductive capacity, either through loss of ancestral worker reproductive capacity to be blocking secondary reductions. Additionally,

because the sting can be used, among other ritualized behaviors, during dominance battles in species with worker reproductive autonomy[44–47], it is expected to indirectly negatively correlate with trophallaxis. In scenario 2 (Eco-opportunism, Fig. 1B), ants that drink sugary liquids acquired from sap-suckers or plants[48,49] need a means to transport these liquids back to the nest, and trophallaxis may have evolved to opportunistically solve this problem when sugary liquid became more abundant in terrestrial ecosystems. Predatory ants typically have a sting and have no need to transport liquids nor to use trophallaxis as prey can be carried. Thus, the shift to a sugary diet may have rendered the sting unnecessary[20]. Species that evolved to exploit a newly abundant resource, such as sugary liquids, may have become capable of sustaining much larger colonies, or because sugary liquids are a less complete diet than insect prey, they may have needed larger colonies to produce new reproductive individuals. In either case, a sugary liquid diet should lead to trophallaxis, loss of sting and large colony sizes. In scenario 3 (social complexity, Fig. 1C), trophallaxis may be an adaptation to high social complexity in large ecologically dominant colonies where resources must be distributed across ever larger groups with potentially different nutritional needs[35,36,42]. In this scenario, other traits bring about the evolution of large colony sizes, only after which does trophallaxis evolve. Finally, in scenario 4 (morphological constraints, Fig. 1D), the sting may not be supportable when ants shift to a low-nitrogen sugar diet and can produce only a thin cuticle[35,50]. In this scenario, reduced intracolonial conflict led to a reduced need for a sting, which then allowed these lineages to take up a sugary diet and acquire trophallaxis.

The best models, according to CICc (Covariance Inflation Criterion corrected for sample size), were the reduction of reproductive conflict and eco-opportunism scenarios where the evolution of trophallaxis was promoted by loss of worker reproductive autonomy and by the integration of sugary liquids in the diet (Fig. 1A, B). Because both models were approximately equally supported, we averaged them (Fig. 1E–G). There was little support for the other two scenarios (Fig. 1E).

Phylogenetic path analysis revealed that a sugary liquid diet both promoted the evolution of trophallaxis behavior (standardized path coefficient 1.06, 95% CI: 0.35–1.78) and trophallaxis promoted a sugary liquid diet (1.17, 0.24–2.10). Trophallaxis, in turn, increased the likelihood of large colony sizes (1.11, 0.11–2.10). Trophallaxis behavior was strongly impeded by the presence of worker reproductive autonomy (−3.43, −6.59−−0.27). Our model did not find a significant impact of any of these traits on the loss of the sting.

These results suggest a causal link between reduction in intracolonial conflict and the evolution of trophallaxis, supporting the hypothesis that trophallaxis should only evolve in superorganismal species. We observe co-occurrence of a diet including sugary liquids and social regurgitation behavior between adults. Whether the dietary shift promoted trophallaxis or trophallaxis existed before and was strengthened by the need to transport sugary liquids remains unclear. Lastly, these results suggest that stomodeal trophallaxis enabled higher social complexity by enabling larger colony sizes.

### Predicting trophallaxis behavior

To analyze the evolution of diet, worker reproductive autonomy and trophallaxis, as well as the contribution of trophallaxis to speciation and extinction dynamics, we needed a larger and more phylogenetically balanced dataset (Fig. S1). We used a Bayesian neural network to train a predictive model able to infer the presence or absence of trophallaxis behavior based on phylogenetic, ecological, and morphological data and life-history traits, which could be collected for a larger number of species. We used the aforementioned traits that correlated with trophallaxis supplemented with proxies for climate preferences on 165 species for which trophallaxis behavior has been reported in the literature. Phylogenetic path analysis was not run on predicted data to avoid circularity.

Our predictive model reached a cross-validation test accuracy of 89% (Table S2). In addition, we asked expert myrmecologists for their

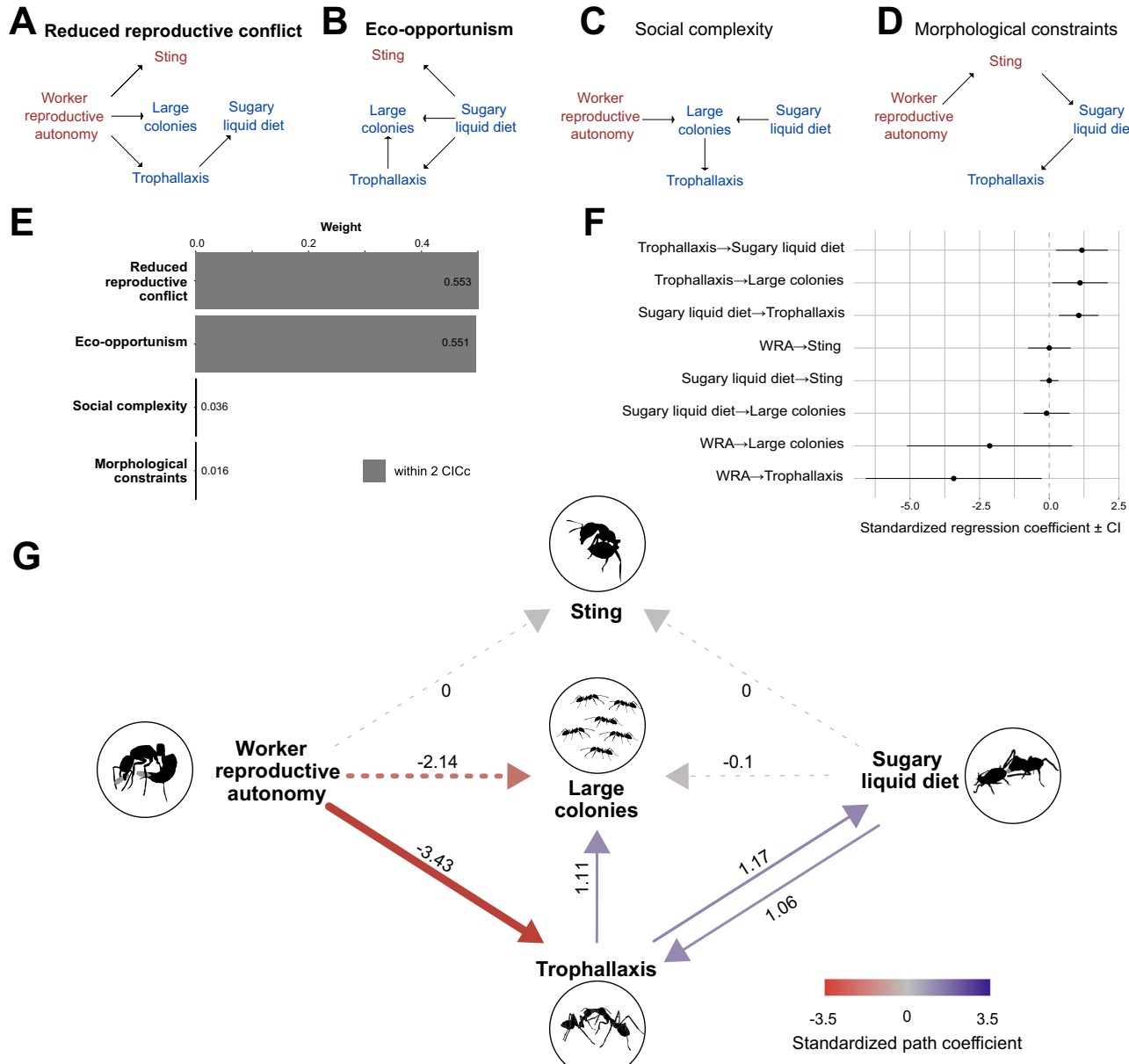

**Fig. 1 | Causal models and results from the phylogenetic path analysis.** Directed acyclic diagram for the **A** Reduced reproductive conflict, **B** Eco-opportunism, **C** Social complexity, and **D** Morphological constraints scenarios. Text in red indicates that the scenario considers the loss of the trait and blue the gain of the trait. Two best models within a ΔCIC <2 are indicated (**E**) and have their titles bolded (**A**, **B**). **F** Representation of 95% confidence intervals for each relationship between traits in the averaged model of (**A**, **B**). WRA worker reproductive autonomy. **G** The averaged model, where dashed lines indicate non-significance (where the confidence interval in panel **F** included zero). Colors correspond to the value of the standardized path coefficient.

assessments of trophallaxis behavior for our predicted species (not used to train the models) and compared experts' assessments of trophallaxis behavior with our predictions (accuracy of 91%, included both false negative and false positive errors, see Table S2), which confirms that our model is unbiased and can accurately predict trophallaxis behavior from this set of traits. Using feature permutation, we identified that the model relies mostly on phylogeny (Δaccuracy of 24.9% ± 0.03). Our model also relies on the inclusion of sugary liquid in the diet (Δaccuracy of 7.8% ± 0.01), worker reproductive status (Δaccuracy of 2.1% ± 0.01), presence or absence of a sting (Δaccuracy of 1.8% ± 0.01) and negligibly, colony size (Δaccuracy of 0.2% ± 0.002) to predict trophallaxis behavior. With our trained model, we predicted trophallaxis behavior for 252 species, more than doubling our taxonomic coverage for this trait and improving taxonomic sampling, representing 15 of the 17 ant subfamilies (Fig. S1).

## The ancestral ant
To understand the evolutionary context around the emergence of social regurgitation between adults, we used phylogenetic comparative methods to assess the concerted evolution of traits, rates of transition, and individual ancestral phenotypes, using a stochastic mapping model in a Bayesian framework. By averaging posterior probabilities over 100 posterior trees, we estimate that the ancestral ant likely had worker reproductive autonomy (posterior probability, pp = 0.63), a sting (pp = 0.98), and small colonies (pp = 0.80), while it was unlikely to rely on sugary liquids (pp = 0.09), to have sterile workers (pp = 0), to have workers that could only lay haploid eggs (pp = 0.37) or to engage in trophallaxis (pp = 0.1) (Fig. S2). From this analysis, we conclude that the profile of the most recent common ancestor of our 417 species resembled ant species from the poneroid clade (Agroecomyrmecinae, Amblyoponinae, Apomyrminae, Paraponerinae, Ponerinae, and Proceratiinae). This result is in line with fossil evidence[51] indicating that,

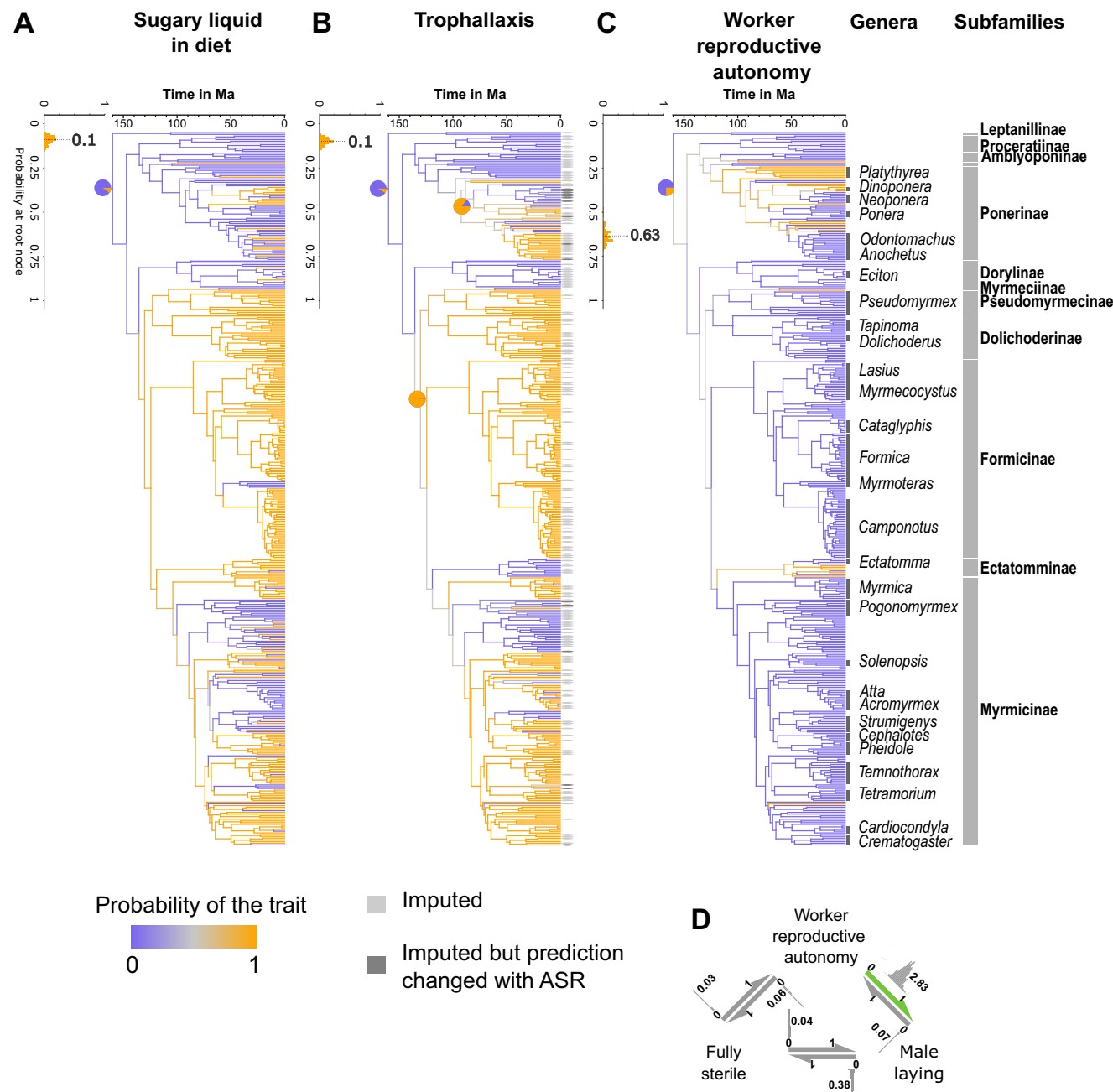

**Fig. 2 | Ancestral state reconstructions for trophallaxis and related traits.**
Ancestral state reconstruction of **A** sugary liquid food in the diet, **B** trophallaxis, and **C** presence of workers with reproductive autonomy on the MCC tree for 417 ant species. The branch color reflects the posterior probability in (**A**, **B**), and the normalized posterior probability of the presence of workers with reproductive autonomy. Pie charts indicate the probability of the trait to be present at a given node based on Log Bayes factors. In tree **B**, we highlight whether the trophallaxis behavior of extant species are imputed (light gray). sMap also recomputed imputed trophallaxis behaviors, and in

dark gray are marked the 17 species in which imputation changed. We highlighted the major genera in our dataset and the major subfamilies. The histogram at the root node of each tree indicates the distribution of the posterior probabilities for the presence of the trait at the root node on the 100 posterior trees. **D** The final averaged model with rates distribution for transitions within worker's reproductive autonomy levels, the green arrow indicating the only transition with rates notably different from 0.

even if early ants had winged queens, it did not preclude the presence of workers with full reproductive autonomy[16].

To compare their evolutionary histories in more detail, we ran Bayesian models on the maximum clade credibility (MCC) tree for each of our traits and averaged them. We had three important nodes whose posterior probabilities remained ambiguous on the MCC tree: (1) The root node for worker reproductive autonomy (pp = 0.54); (2) the most recent common ancestor of the non-doryline formicoids for trophallaxis (pp = 0.53); and (3)

the most recent ancestor of *Centromyrmex* and *Odontomachus* in our dataset for trophallaxis (pp = 0.49), referred to as the late Ponerini node. To clarify these ancestral states, we fixed the root node of the subtree to either state and compared marginal likelihoods using Bayes factors[52], using both "all rates differ" and "equal rates" models (Table S5), and then computed the relative model probability of the node to have either state. For reproductive autonomy, this indicated that the root node had only a 25% relative model probability to have workers with full reproductive autonomy and 75% to

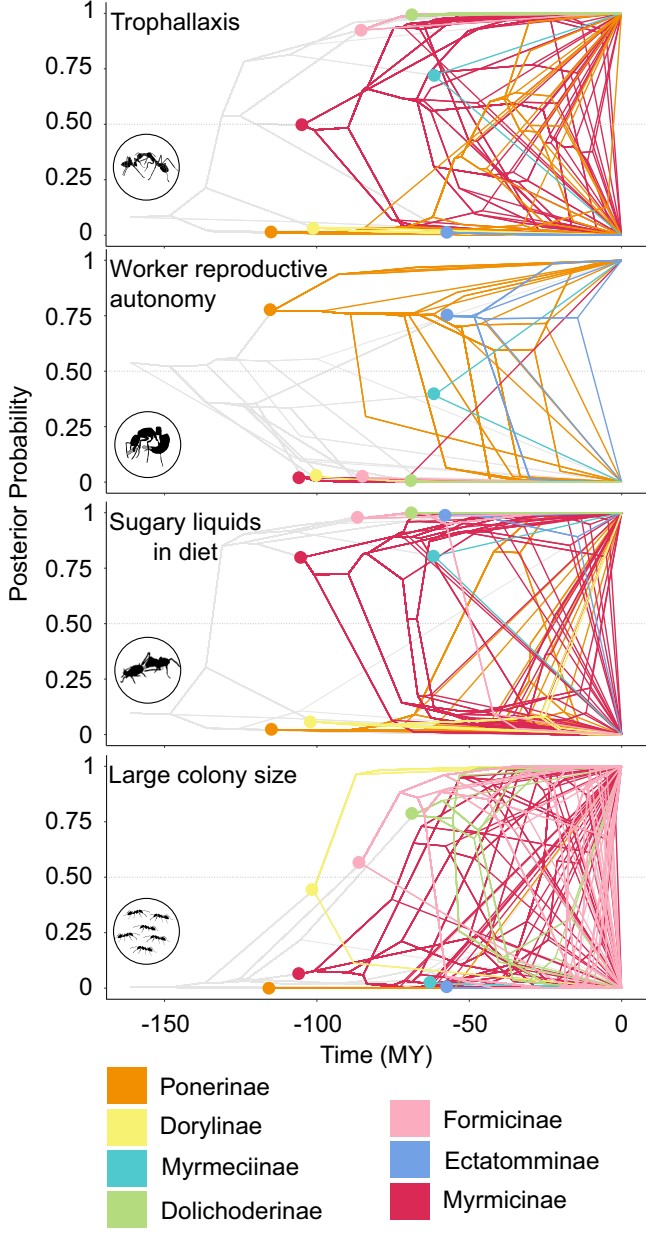

**Fig. 3 | Phenograms of the posterior probabilities for trophallaxis, worker reproductive autonomy, sugary liquids in the diet, and large colony size, for species present in our full dataset.** The circles identify the most recent common ancestor of each subfamily. For traits with more than two states (Worker reproductive autonomy and Colony size), we illustrate the normalized posterior probability of the state of worker reproductive autonomy and large colony size, respectively.

have workers whose reproductive autonomy had already been limited to only laying male eggs (pie charts in Fig. 2 and Table S5). The same analysis supported the conclusion that the first non-doryline formicoid node had already evolved trophallaxis with a relative model probability of 99% and that the late Ponerini node had already evolved trophallaxis with a relative model probability of 85%.

To reconcile these analyses regarding worker reproductive autonomy on the MCC tree (relative model probability = 0.25, Fig. 2C node) and the ancestral state reconstructions on 100 posterior trees (pp = 0.63, Fig. 2C inset), we looked at transition rates. The rates of our averaged Bayesian model indicated that worker reproductive autonomy is more easily lost than gained with regard to both workers that can lay only male eggs and completely sterile workers (Fig. 2D and Table S3), with rates indicating a nearly

irreversible reduction in worker reproductive potential. Generally, the near irreversibility of worker reproductive potential is consistent with literature on the entrenchment of the division of labor in major evolutionary transitions[11,19,40,53,54], and supports the idea that the ancestral ant in our tree was not yet superorganismal. Beyond the root, many major nodes early in the ant phylogeny remain ambiguous regarding worker reproductive autonomy (e.g., pp = 0.36 at the root node of formicoids), indicating the clear need for further study and also highlighting a few clearly credible secondary reductions (the lineage leading to *Metapone madagascarica* around 35 Ma, and to *Myrmecia pyriformis* around 50 Ma).

A diet including sugary liquids likely evolved independently at least 13 times across our tree (Figs. 2, 3), a major gain about 130 Ma, one time in the lineage leading to Paraponerinae, around 60 Ma, nine times in lineages leading to 18 ponerine species (between 60 and 10 Ma) in genera *Platythyrea*, *Dinoponera*, *Pachycondyla*, *Neoponera*, *Diacamma*, *Mesoponera*, and *Odontomachus*, and twice in the genus *Labidus* (around 15 Ma). A vast majority of these ponerine gains are in opportunistic sugary liquid feeders that remain primarily predatory. The major gain of sugary liquids in the ant diet occurred at the most recent common ancestor of non-doryline formicoids (Myrmicinae, Ectatomminae, Formicinae, Dolichoderinae, Aneuretinae, Myrmeciinae, and Pseudomyrmecinae), between 140 and 130 Ma, possibly coinciding with major increases in angiosperm biodiversity and abundance[55]. Dating these two phylogenies is a highly challenging task due to the size of the clades, the relative sparsity of their fossil records, and given that the time of origin of flowering plants remains hotly debated. Yet, based on available data, the first transition to trophallaxis corresponds with phases of angiosperm expansion based on recent molecular clock dating[56] and on fossil-based estimates that infer a peak in angiosperm family-level diversification rates in the early Cretaceous, around 130 Ma[55]. These subfamilies encompass most major tenders of sap-sucking insects and typically display an 'omnivorous' diet heavily reliant on sugary liquids. We observe one clear shift away from sugary liquids in the formicine genus *Myrmoteras* (Formicinae) around 50 Ma and multiple shifts away from sugary liquids in the Myrmicinae clades *Pogonomyrmecini*, *Stenammini*, *Attini*, *Crematogastrini*, *Solenopsidini*, and *Myrmicini* over the last 95 Ma, and in *Nothomyrmecia* (Myrmeciinae) around 40 Ma. The model indicates that it is slightly easier to begin drinking sugary liquids ($r = 1.32$, 95% CI 1.30–1.34) than to stop ($r = 1.02$, 95% CI 1.01–1.04). However, these rates are in the same order of magnitude, indicating that it is a plastic and volatile trait.

Our ancestral state reconstruction of trophallaxis revealed that the behavior evolved and was lost multiple times, with two large gains about 90 Ma in Ponerinae and 130 Ma in non-doryline formicoids (Figs. 2, 3). Trophallaxis was gained in *Ponerini* with a relative model probability of 85% about 90 Ma and led to 29 extant species in nine genera predicted (22) or known (7) to use trophallaxis. However, in ponerines, trophallaxis is rarely observed and is opportunistically related to food; more data on this clade, particularly in *Odontomachus* and *Anochetus*, would allow confirmation of these predictions. We did not find any record of species using stomodeal trophallaxis in dorylines, though a once-in-a-lifetime social fluid exchange was recently described in *Ooceraea biroi* between pupae and adults[57]. The challenging nature of keeping many Dorylinae species in the lab poses major obstacles to the observation of individual behavior. Therefore, it remains uncertain whether some might engage in stomodeal trophallaxis.

The major gain of ant trophallaxis occurred in non-doryline formicoids around 130 Ma. We determined that the ancestor of non-doryline formicoids in our tree had a 98% relative model probability to have evolved trophallaxis, indicating one single gain of trophallaxis at this node. We identified several losses of trophallaxis in *Nothomyrmecia macrops* (Myrmeciinae) between 50 Ma and 40 Ma, at the origin of Ectatomminae around 120 Ma, in Myrmicine clades *Pogonomyrmecini* (100 Ma), *Stenammini* (90 Ma), some *Attini* (between 70 and 10 Ma) and in *Metapone madagascarica* (Myrmicinae) (around 70 Ma). This single gain is well-supported, with clear evidence of its presence in four distinct clades: (1) the subfamilies Dolichoderinae, Aneuretinae, Pseudomyrmeciinae, and Myrmeciinae; (2)

the Formicinae subfamily; (3) the Myrmicini tribe within Myrmicinae; and (4) the tribes Solenopsidini, Attini, and Crematogastrini, also within Myrmicinae. In particular, in Myrmicinae, trophallaxis has been lost several times, making it hard to identify clear evolutionary paths, but also suggesting that the trait is highly plastic. The averaged Bayesian model shows that trophallaxis is slightly easier lost ($r = 0.70$, 95% CI: 0.69–0.72) than gained ($r = 0.51$, 95% CI: 0.50–0.53) (Table S3), but the rates being of similar magnitude indicates that the trait is labile. Thus, our results suggest that ants' diet is a plastic trait, which is consistent with the hypothesis that trophallaxis evolved as a means to exchange sugary liquids.

We also reconstructed the evolution of colony size across our tree (Fig. 3). Our analysis indicates that large colony sizes first evolved around 100 Ma in the most recent common ancestor of the genera *Labidus*, *Nomamyrmex*, *Eciton*, *Neivamyrmex*, and *Aenictus* (Dorylinae) in our dataset. We also observe the evolution of large colony sizes in nodes preceding our formicine species (around 85 Ma) and dolichoderine species (around 70 Ma), both gains being followed by multiple shifts back to medium or small colony sizes. In ponerines, only the lineage leading to *Brachyponera chinensis* evolved large colonies (around 30 Ma). In myrmicines however, many lineages leading to extant species evolved large colony sizes starting around 70 Ma, sometimes at the genus level (*Atta*, *Acromyrmex*, *Tetramorium*), and sometimes for individual species (*Crematogaster ashmeadi, Myrmica rubra*). The rates of the model indicate that shifts between small and large colony sizes ($r < 0.1$) are rare, in agreement with previous work[58], and that shifts between small and medium ($r = 3.94$, 95% CI 3.91–3.97) and medium and large ($r = 4.09$, 95% CI 4.06–4.12) colonies are more likely, with similar rates in both directions. The fact that these shifts are reversible may partially reflect noisy trait data, as it is difficult to obtain consistent colony size data over many species where it is not possible to correct for colony age or sampling biases.

Put together, these results highlight one major evolution of trophallaxis and multiple subsequent gains and losses, illustrating some level of plasticity in this social regurgitation behavior. We note a clear link between the shift in diet from predaceous to relying more on sugary liquids between 140 and 130 Ma, that overlaps with the early phases of Angiosperm diversification[59–63] and associated sap-sucking insects[64,65]. While there remains uncertainty around the worker reproductive autonomy of ancestral ants, combining the path analysis (Fig. 1) with the near irreversibility of reduction in worker reproductive potential (Fig. 2D) and the known bias against worker reproductive autonomy in the dataset (see Methods) lead us to conclude that the most parsimonious explanation is that the ancestral ant worker maintained reproductive autonomy.

### Conditions the evolution of trophallaxis

Analyzing the sequence of trait evolution in different ant lineages, a pattern emerges for the conditions that allowed this social transfer to evolve. The pattern of evolved traits observed across the majority of our ant species was the loss of worker reproductive autonomy, the early (140–130 Ma) integration of sugary liquid in the diet, and the early evolution of trophallaxis (starting around 130 Ma) (Figs. 2, 3). This happened at the most recent common ancestor to the non-doryline formicoids of our dataset, which impacted seven subfamilies (Pseudomyrmecinae, Dolichoderinae, Aneuretinae, Myrmeciinae, Formicinae, Ectatomminae, and Myrmicinae), accounting for 86% of all ant species. This leads us to the conclusion that trophallaxis evolved given the pressure of ecological opportunism to share liquid food and only when intracolonial reproductive conflict was sufficiently low.

The ingestion of liquid food alone appears insufficient to lead to the evolution of trophallaxis, as seen in some *Platythyrea* or Ectatomminae. Multiple other subfamilies (Leptanillinae, Apomyrminae, Proceratiinae, and Agroecomyrmecinae, Fig. 2) never integrated sugary liquids into their diet, workers of extant species do not have reproductive autonomy, and trophallaxis never evolved. This is consistent with the phylogenetic path analysis and indicates that low intracolonial conflict alone was insufficient to drive the evolution of trophallaxis.

The sequence of trait evolution surrounding the gain of trophallaxis in the late *Ponerini* remains cloudy. The evidence for this gain remains somewhat weak because it relies heavily on imputed behavioral data: 7/55 ponerine species in our dataset are known to use trophallaxis, and 22/55 have been imputed to use trophallaxis, likely due to the fact that they drink sugary liquids opportunistically. Many ponerines are reported as purely predaceous but still drink sugary liquids opportunistically, so trait classifications may change with further study.

The only extant exceptions to our pattern in Ponerinae are *Diacamma* species (*Diacamma rugosum* in our dataset and *Diacamma cf. indicum*, not present on this tree) that have gamergates[66] and are either predicted or known to use trophallaxis. However, workers of these species are castrated late in their development when in the presence of a reproductive female[67,68], making them incapable of reproductive autonomy as adults. Thus, reproductive conflict was likely sufficiently low to allow trophallaxis to evolve.

In myrmicines, there are many genera and species whose trait evolution is consistent with our model (e.g., *Cephalotes*, *Tetramorium*, *Acanthomyrmex*, and *Nesomyrmex*). However, we also have 31 Myrmicinae species not known to drink sugary liquids but known or predicted to use trophallaxis, such as *Pheidole bicornis* or *Cardiocondyla nuda* (Figs. 2 and S3). We observe one clade in formicines that consistently uses trophallaxis while not drinking sugary liquids (*Myrmoteras*). These examples indicate that trophallaxis behavior may be worth maintaining even once the pressure for liquid food sharing has been removed. One species, from the subfamily Myrmeciinae, *Myrmecia pyriformis* is a clear exception to our pattern, in that they have reproductive workers, do not drink sugary liquids and use trophallaxis[69].

Finally, there is no clear pattern regarding the evolution of large colonies. In some species it evolved early, for example in some dorylines (around 100 Ma), a lineage that never evolved trophallaxis, while many Myrmicinae species evolved trophallaxis but never evolved to have large colonies (*Temnothorax*).

Overall, with the exception of *Myrmecia pyriformis*, all extant species are consistent with the model that trophallaxis evolved to opportunistically exploit ecologically abundant liquid foods once intracolonial conflict over reproduction was contained through reduction of worker reproductive autonomy.

### Speciation and extinction analyses

Given that some of the most species-rich lineages of ants engage in trophallaxis, we tested whether trophallaxis could have impacted speciation and extinction patterns over the ant phylogeny. Using a Bayesian Analysis of Macroevolutionary Mixtures (BAMM)[70] with STructured Rate Permutations on Phylogenies test (STRAPP)[71], we found that trophallaxis led to significantly increased net diversification ($p = 0.027$), with lineages engaging in trophallaxis showing 1.7× higher speciation rates (no trophallaxis = $0.068 \pm 0.004$; trophallaxis = $0.116 \pm 0.007$) and 3.8× higher extinction rates (no trophallaxis = $0.005 \pm 0.004$; trophallaxis = $0.019 \pm 0.007$). When accounting for the possibility of other, unmeasured traits affecting diversification using a state-dependent model with hidden states (HiSSE)[72], we found that the preferred model was for trophallaxis-dependent diversification with two hidden states (delta AIC = $-372$; Table S4). Thus, the HiSSE model also supports a link between trophallaxis and increased net diversification (Fig. 4). These results are consistent with the omnipresence of ants that use trophallaxis and the maintenance of trophallaxis behavior in lineages that have ceased to drink sugary liquids.

### Discussion

Social regurgitation is a plastic behavior that evolved in ants to share liquid food cooperatively, but only in colonies with sufficiently reduced reproductive conflict i.e., restriction of worker reproductive autonomy. Given that trophallaxis and worker reproductive capacity are indicators of colony cooperation, conflict, and reproductive autonomy, this behavior is an excellent marker of superorganismality. It exemplifies the intricate social interactions, cooperative resource allocation, reproductive division of labor,

**Fig. 4 | Trophallaxis increased net diversification.** Representation of the character-dependent HiSSE model with two hidden states. q indicates transition rates between state combinations, μ is extinction, λ is speciation, net diversification rate r is calculated as $r = \lambda - \mu$, ε is the extinction fraction, calculated as $\varepsilon = \mu/\lambda$. For each state, the extinction fraction was 0.132. Red indicates trophallaxis and the absence or presence of a gray ring indicates the hidden state.

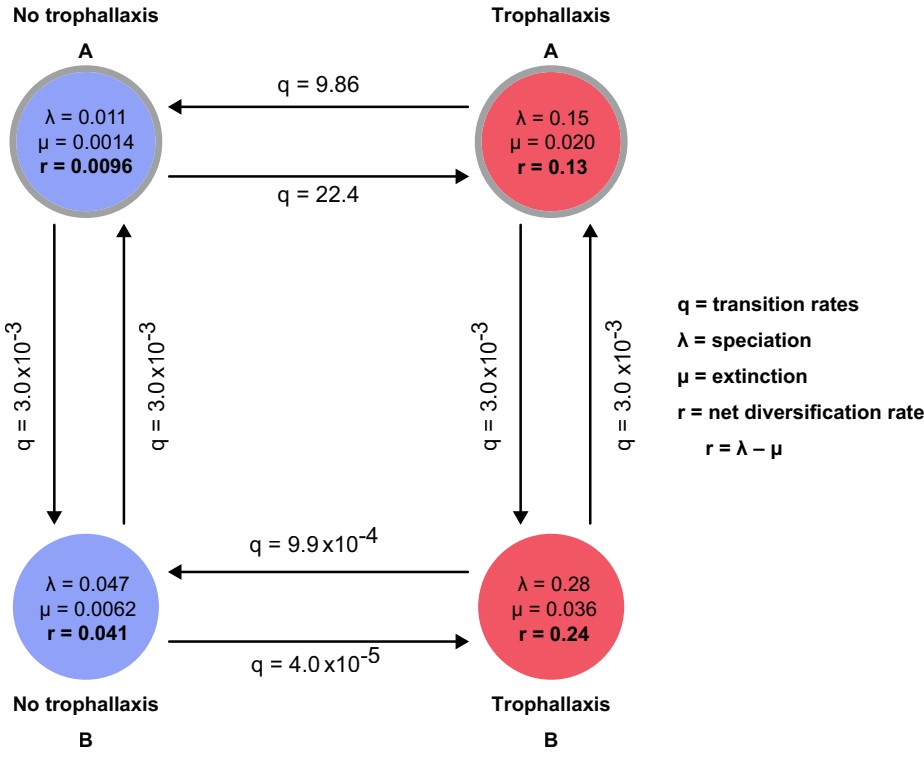

and metabolic division of labor that define higher-order structures following major evolutionary transitions in individuality.

The high number of gains and losses of trophallaxis, especially in myrmicines, as well as the similar rates of our evolutionary model for trophallaxis, support the idea that trophallaxis is a plastic behavior that could have evolved as an adaptation to crucial shifts in ecological niches and diets. The integration of sugary liquids into the ant diet may have been promoted by the two global increases in angiosperm diversity and abundance[56,59,61,62], though merging the dating of these separate phylogenies should be the topic of future work. The integration of sugary liquids into the ant diet represents a key shift from a purely predaceous diet to a generalist diet (including both prey and sugary liquids), which in other systems, has been linked to faster diversification[73,74], range expansion[75,76] and invasion[77–80], also in ants[60]. Despite clear advantages to becoming a generalist, not all ants did so when sugary liquids became abundant, and the relatively high degree of phylogenetic conservatism in the distribution of presence or absence of trophallaxis, suggests that the behavior relies on clade-specific morphological features (e.g., proventriculus and musculature). Trophallaxis is plastic also in the way it is used, with various frequencies of events across the ant phylogeny, from observations of trophallaxis only after release from starvation, in some ponerines, or very frequent interactions outside the context of food in some formicines. This, together with the two main gains of trophallaxis we've identified, beg the question of whether the two behaviors should be considered the same. Trophallaxis likely first evolved in ants as a mode of sharing exogenous food, yet in ants that perform this behavior frequently (e.g., *Camponotus*), trophallactic fluid is a rich and complex socially transferred material that can alter larval development in a parallel to mammalian milk[21–23].

The importance of colony size in the evolutionary history of trophallaxis remains cryptic because this trait is linked with other aspects of colony organization, especially with worker reproductive autonomy. Larger colony sizes have been proposed to increase reproductive inequalities between reproductive individuals and workers—as larvae of large colonies have less chance to become a new reproductive—but also between workers, with increased worker policing[42]. Our results indicate that trophallaxis, perhaps through enabling metabolic division of labor and/or a tailored larval

diet[36], could have allowed the production of cheaper workers[35] and great control of queen:worker ratios. Thus, social complexity, colony size, and the social circulatory system are tightly linked together to participate in the ecological dominance of species that use trophallaxis.

Whether workers of the most recent common ancestor to all living ants maintained worker reproductive autonomy remains unclear. However, our analyses suggest a near-irreversible reduction of worker reproductive autonomy and that the reduction of reproductive autonomy strongly promoted the evolution of trophallaxis. These results stand despite a strong bias in our dataset against an ancestral state of worker reproductive autonomy. Ants are generally assumed to have workers that cannot be mated and can only lay haploid eggs, yet there are species with full worker reproductive autonomy or complete worker sterility. Here, we annotated a species' worker reproductive capacity as either known to have gamergates (maintaining worker reproductive autonomy with an intact spermatheca, capable of laying diploid eggs), known to have sterile workers (lacking both ovaries and spermatheca), or the remaining species (in many cases, unstudied, but typically can only lay haploid male eggs). This assumes that most species, where worker reproductive capacity has not yet been investigated, have reduced reproductive autonomy, biasing ancestral state reconstructions. Together, these suggest that the most recent common ancestor to all living ants in our dataset likely maintained full worker reproductive autonomy. More work is necessary to tackle this uncertainty through a more thorough assessment of more species' worker reproductive capacity, both on extant species and on fossilized ants. If this result holds, it suggests that not all ants are superorganismal, and that superorganismality may have evolved multiple times within Formicidae. This would allow researchers to view ant diversity in a new light, allowing us to contrast different evolutionary origins of extreme social complexity.

Our scenario for the evolution of social regurgitation between adults in ants differs from the hypothesized evolutionary paths for trophallaxis in wasps, bees, and termites[34,81–83]. In wasps, trophallaxis likely evolved from adults pre-chewing prey and providing it to larval young, and generally, adult-adult trophallaxis only occurs in highly eusocial wasps[81]. In bees, trophallaxis also likely first arose in the context of nutrient provisioning for young[82], but has also been linked to reproductive conflict. Indeed, in several

bee species, asymmetrical trophallactic relationships are key to reproductive dominance hierarchies[28,34]. In termites, parent-to-offspring proctodeal trophallaxis was a key evolutionary innovation for their symbiont-enabled lifestyle[4,83]. None of these systems have been evaluated in the phylogenetic detail we have done here.

Given that trophallaxis evolved in each of these groups in connection to parental care, it highlights the unusual case of the ants. In ants, because the data on larval-adult trophallaxis are so sparse[20], we were not able to explore the possibility that another social transfer may have been the precursor in the evolution of stomodeal trophallaxis between adults. That said, most predatory ants do not feed larvae in individualized interactions like trophallaxis[84], and we and others clearly find that the common ancestor to all ants was predatory[32,85]. Pupal molting fluid secretion to ant workers[57] is another social transfer within the colony that likely contributed to early interdependencies between castes that may have strengthened cooperation in the colony, but because it is only a one-way transfer and one-time event per donor individual, it is not sufficient to build a functional social circulatory system across a colony[36].

Alternative modes of nutrient exchange, such as feeding on larval hemolymph through a larval hemolymph tap, tubercle, or puncture, or producing trophic eggs, could serve a similar purpose as trophallaxis in creating a network of metabolic division of labor across the colony[20,36]. However, it remains difficult to test this hypothesis due to the scarcity of data, especially in the annotation of species that do not use these modes of nutrient transfer. When these sparsely annotated traits are viewed in the context of trophallaxis evolution (Supplementary Fig. S8, from data in ref. 37 and our social exchanges database[86] extrapolated to genus level), there is no clear interaction between the use of trophic eggs and trophallaxis, but larval hemolymph feeding is only found to occur in species not using trophallaxis. This could mean that larval hemolymph feeding might be an ancestral nutrient exchange behavior and/or a behavior secondarily evolved in genera that lost trophallaxis. While we found no clear correlation between trophic eggs and trophallaxis, this could be due to true biology or insufficiently precise trait annotation. Indeed, there are several species of Dolichoderinae and some Myrmicinae that are highly reliant on trophic eggs that would phylogenetically be expected to perform trophallaxis but do not[87,88]. This is a fascinating domain for future study.

Studying the macroevolutionary history of behavior and its consequences on evolution is challenging because we cannot easily measure behavior from fossil evidence, and some behaviors can be difficult to assess even in extant species. Here we have adapted new deep-learning tools to impute behavior, allowing us to bridge this gap and study the evolution of a complex behavior over a large phylogeny. Bayesian neural networks provide a flexible tool to integrate multiple traits along with phylogenetic structure in our predictions, while embracing and explicitly quantifying the uncertainties around them, but they can only be trusted as much as the input data can be. With this, performing a similar analysis using a more recent phylogeny would be worthwhile. In our analytical framework, these uncertainties were then propagated to the comparative phylogenetic analysis and ancestral state estimations to provide robust inferences. While such machine learning tools are valuable, further behavioral annotations in more species can only improve our understanding of behavioral evolution, and we encourage scientists to report observations of behaviors like trophallaxis. The timing of the major shifts we inferred in ant behavior and diet coincide with the two major moments of diversification of flowering plants in most terrestrial ecosystems[56,63]. Our results suggest that this simple but important behavior had a notable impact on net diversification and, thus, the evolution of different ant lineages. Fundamentally the shift to trophallaxis was enabled by ant-plant and ant-sap-sucker interactions. The adoption of trophallaxis resulted in cascading adaptations across multiple partners, and thus significant rewiring of ecosystems[3,32]. Ants, and in particular ants that rely on such mutualisms, are known to be ecosystem engineers, moving their aphid cattle from plant to plant, protecting trees from predators, cultivating fungus gardens, and dispersing seeds. Understanding the evolutionary underpinnings of behavioral traits is crucial to grasp their fundamental role in determining the ecological success of the most dominant organisms in the animal kingdom.

## Methods
### Data collection
We completed the reports of trophallaxis from ref. 20 by manually searching the literature up to early 2021 for reports of occurrence or absence of trophallaxis behavior, obtaining data for trophallaxis behavior of 205 species from 265 sources[86]. We searched Google Scholar for combinations of genus and/or species with "trophallaxis", "oecotrophobiosis", "regurgitation", and "share liquid", which are means by which trophallaxis has been described in scientific literature. We considered that a species does not use trophallaxis if it was specifically mentioned in the literature that trophallaxis was never observed. Because mentions of the absence of behavior are less frequent than mentions of the behavior, because some subfamilies were more represented than others, and to have more even taxon sampling, we imputed trophallaxis behavior for additional species (Fig. S1). To do so, we used a Bayesian neural network (BNN) as implemented in the npBNN library[89] (https://github.com/dsilvestro/npBNN), that provided posterior probabilities for a binary trait, based on life-history, morphological and ecological traits collected from literature. These traits were either easy to collect for a large number of species (temperature and humidity data, presence or absence of a sting) or were linked to some of our hypotheses. We acknowledge that because of information availability on trophallaxis behavior and other traits, our sampled dataset is not random. However, the dataset with imputed trophallaxis behaviors included species in 15 subfamilies, with 2–6% of species covered across all subfamilies with >2 species (Fig. S1). See supplementary code for more details[90]. Our core dataset (known, not imputed) lacks representative species of early branching subfamilies (Leptanillinae and Martialinae) due to a lack of behavioral data and the rarity of these ants (Fig. S3).

### Phylogeny
We used the maximum clade credibility (MCC) tree produced by ref. 91 as a reference for the ant phylogeny of over 14,000 species, as it was the largest and most recent ant phylogeny at the time. This phylogeny was built with genetic information for 673 species, 86 of which are in our dataset, representing a total of 20.7% of our species. We corrected species names according to antwiki valid species names in 2021 (https://www.antwiki.org/wiki/Ant_Names) and removed morphospecies. To include the phylogenetic signal in the BNN to predict trophallaxis behavior, we decomposed the tree into eigenvalues using the R packages ape v5.4.1[92] and PVR v0.3[93], providing us with 11 eigenvalues (making up 50% of the phylogenetic signal) per species that we used as an input to the BNN. Data are available in the associated repository[90].

### Temperature and humidity
We retrieved temperature and humidity data from the GABI database (Release 18.01.2020) and Antweb (October 2020). Temperature and precipitation data were obtained from the coordinates referenced in the two aforementioned databases, using the R package raster v3.3.13. For each specimen of each species, we collected the mean annual temperature and precipitation recorded at this location. For each species, we then extracted a range of mean annual temperatures and precipitations, thus providing a range of temperatures and precipitations for each species' habitat, ending in four continuous traits: minimum and maximum temperature and minimum and maximum precipitation. Data are available in the associated repository[90].

### Drinking sugary liquids
Mention of a species opportunistically drinking sugary liquids in the field, tending aphids, or having repletes was reported as 1, and specialized predators or ants not interested in sugary foods were attributed a 0. For four species, we accepted information indicating species drinking sugary liquids in the lab (*Strumigenys lewisi*, *S. canina*, *Cephalotes varians*, and *Streblognathus aethiopicus*). For 32 species in 17 genera (*Acanthognathus*, *Anochetus*, *Apomyrma*, *Cerapachys*, *Cryptopone*, *Discothyrea*, *Leptogenys*, *Myopias*, *Myrmecina*, *Neivamyrmex*, *Platythyrea*, *Ponera*, *Prionopelta*, *Probolomyrmex*, *Proceratium*, *Typhlomyrmex*, and *Tapinoma*), the species-level diet information was impossible to retrieve, so it was inferred from the genus-level.

## Sting

Whether species have a functional sting or not was retrieved from the supplementary material of ref. 94 and extrapolated from genus to species level. Missing values in the training set were retrieved manually from scientific literature.

## Colony size

Colony size data were retrieved manually from scientific literature, AntWiki.org, and online shops specialized in ants when not found in other sources. We searched Google Scholar and Google using a combination of the genus and/or species with "colony size" and "colony" as keywords. Because colony size is variably reported in the literature (sometimes maximum, minimum, average, or single observations), we recorded what was available. To face this heterogeneity in data collection, we split colony size into three categories (small: 0 to 279, medium: 280 to 4999, large: 5000+).

## Worker reproductive autonomy

Categorical data on full worker reproductive autonomy (ants with worker fate but that can be mated and produce diploid offspring, thus that have functional ovaries and spermatheca[10]) was retrieved from antwiki, https://www.antwiki.org/wiki/Category:Gamergate (October 2021) and confirmed with literature search[16,95]. Genera not listed were assumed to have workers that could not be mated (non-gamergates). Species from genera known to have gamergates but for which no data were found were not included in the dataset, as we do not know how labile the presence or absence of gamergates is within a genus.

We also collected data on sterile workers unable to lay even haploid eggs[10] at the genus level and extended to the species level. However, in genera when worker sterility is labile across species, we did not infer worker sterility from the genus level and searched for data on specific species. Finally, by default, species in our dataset not known to have gamergates or completely sterile workers, are assumed to have workers that can lay haploid eggs. This group contains twelve species[96] which have either workers (e.g., *Ooceraea biroi* or *Platythyrea punctata*) or queens (e.g., *Paratrechina longicornis* or *Wasmannia auropunctata*) able to reproduce asexually. While this 'clonal' reproduction offers a profound degree of reproductive autonomy, clonality in each of these cases is thought to have evolved relatively recently, is rare in some species, is only found in certain populations of others, and was not followed by diversifications. Consequently, for simplification, we treated these twelve species as having non-gamergate/non-sterile haploid-laying workers.

## Data imputation

After collecting data, we had 163 species for which we knew all traits. We wanted to expand and improve the phylogenetic sampling of our dataset. We trained a Bayesian neural network (npBNN, v0.1.15[89]) on these 163 species split as training (90%) and test sets (10%). To evaluate the test accuracy of the model, we performed a tenfold cross-validation analysis. The BNN model included a fully connected network with two hidden layers of five and five nodes, respectively, with a "tanh" activation function[97]. We used a normal prior. The final model was trained on all 163 data.

The cross-validation test accuracy of our model was 89% on average over the ten test samples. On average, 11.7 species were classified correctly as using trophallaxis (True Positives), 2.5 species were classified correctly as not using trophallaxis (True Negatives), 0.8 species were classified incorrectly as using trophallaxis (False Positive), and one species was classified incorrectly as not using trophallaxis (False Negative) (Table S2). The averaged accuracy, precision, recall, and F1-score of the cross-validations indicate that our model does not overfit.

After running the trained model on 252 species where we did not know trophallaxis behavior, we obtained a dataset of 417 species: 163 species with known trophallaxis behavior and all known traits, two species with known trophallaxis behavior but missing temperature and precipitation data (*Myrmicaria natalensis eumenoides*, and *Myrmoteras jaitrongi*). This dataset is available in the supplementary repository[90].

To further evaluate our model, and to verify it does not overfit with an independent test set, we sent the list of species for which we predicted trophallaxis behavior to ten expert myrmecologists (C. Lebas, N. Idogawa, A. Yamada, P. Slingsby, R. Mizuno, J. Heinze, A. Lenoir, V. Nehring, F. Savarit, and X. Cerda) and asked them if they had observed or had not observed stomodeal trophallaxis between adults for any species on the list. Experts provided responses for 47 species, 18.7% of our imputed species. We have 38 true positives, 5 true negatives, 3 false positives, and 1 false negative, leading to a model accuracy of 91% on myrmecologists answers (Table S2), confirming that we can reasonably trust our imputations. When our predictions were false, we corrected according to experts' answer in later analyses.

Our model allowed us to predict trophallaxis behavior for 252 species. For almost all subfamilies, except Martialinae, Myrmeciinae, and Aneuretinae (only one species in the subfamily, with known trophallaxis behavior), we managed to predict the trophallaxis behavior of additional species, thus expanding our dataset. Before imputation, the percentage of species in a subfamily with more than two described species (all except Martialinae, Agroecomyrmecinae, Aneuretinae, Apomyrminae, and Paraponerinae) with known trophallaxis behavior ranged from 0.3 to 3.4%. After imputation, these species represent between 2.1 and 6% of the subfamilies, confirming that we managed to reduce the taxon-sampling bias due to species diversity between subfamilies (Fig. S1).

## Evolutionary analyses

We used sMap to infer ancestral states using phylogenetic comparative methods. For each trait (presence or absence of workers that can mate, sting, trophallaxis and sugary liquid food in species diet; discretized colony size) we evaluated different Markov models in a maximum likelihood framework and ranked them based on Akaike weights. Analyses were run with 1000 simulations of trait evolution using the "sMap" command. We ran maximum likelihood analyses on both the maximum clade credibility tree and 100 posterior trees to take phylogenetic uncertainty into account, when determining the status of each trait at the root node.

Further analysis was done only on the maximum clade credibility tree. We built a Bayesian model with a gamma prior (shape parameter $k = 2$ and scale parameter $q = 2$) and computed the posterior probability of each model and used it to average Bayesian models with the "Blend-sMap" command. Because sMap allows the probability of a tip state as an input, we used posterior probabilities from the BNN imputation of trophallaxis behavior in our analyses, and sMap uses them as priors. Twenty-one trophallaxis imputations were reassessed by sMap resulting in a change in the probability of trophallaxis behavior (shown in Fig. 2B). Out of 21, four had a trophallaxis imputation between 40% (uncertain absence of trophallaxis) and 60% (uncertain presence of trophallaxis).

## Trait correlation and phylogenetic path analysis

We used a d-test[39] as implemented in sMap[98] (v1.0.7) to estimate the co-occurrence of each trait with trophallaxis. We performed this test on the dataset of species with known and expert-verified trophallaxis behavior (212 species) to avoid circularity, and separately on the full dataset containing imputations to observe whether there was any difference (Table S1). We sampled 100 histories from the posterior predictive distribution of each parameter sample. Then, we averaged Bayesian models and pairwise merged the trophallaxis model with every trait using the "Merge-sMap" command to finally use "Stat-sMap" with default values to run the D-test.

To test how traits influenced trophallaxis behavior, we used phylogenetic path analysis as proposed by ref. 41, on the dataset of species with known and expert-verified trophallaxis behavior only (212 sp.) to avoid circularity. This method allows the comparison of possible causal relationships between traits while testing for direct or indirect effects and considering the non-independence of the traits due to phylogeny. We first built causal quantitative models as directed acyclic graphs (DAGs, Fig. 1A–D) and accounted for phylogeny through phylogenetic generalized least squares analysis (PGLS), using the d-separation method[41,99]. Building

the models, running the phylogenetic path analysis, and performing model selection was done in R with the Phylopath v1.1.3[100], with 500 bootstrap replicates to take both continuous and discrete response variables into account. The four models were built to describe possible scenarios of the evolution of trophallaxis when considering our traits of interest.

The best models were respectively "reduced reproductive conflict (Fig. 1A) and "eco-opportunism" (Fig. 1B) ($p$ value >0.05 and DCICc <2, Fig. 1). These two models were averaged using the average() function in the phylopath package. This provided a model that reflects the results of the best models (Fig. 1E–G) which rejects the "social complexity" (Fig. 1C) and "morphological constrains" models (Fig. 1D).

### Speciation and extinction analyses

We used reconstructed birth-death processes to estimate the dynamics of speciation and extinction rates across the ant phylogeny and assess whether the evolution of trophallaxis might have affected species diversification. To this end, we used the state-dependent models implemented in the R package HiSSE (v1.9.6)[72] and the mixture models implemented in BAMM (v2.5.0)[101].

For the HiSSE analysis, we used the tree from ref. 91 tree with 14,183 species, and set the trophallaxis state to unknown for all species missing from our imputed dataset, as it would capture phylogenetic information better than through the estimation of the sampling fraction. We tested three models (Table S4), starting with (1) the original BiSSE model[102], with net diversification (speciation - extinction) and extinction fraction (extinction/speciation) dependent on that trait state. Because the BiSSE model is prone to false positives[72], we also ran a character-dependent HiSSE model (2) with one hidden state, where diversification rate variation can occur as a function of both trophallaxis and an unobserved (hidden) trait. We additionally tested a character-independent model (3) with one hidden trait affecting net diversification and extinction fraction and with no effect of trophallaxis. We compared the fit of these models through maximum likelihood using AIC (Table S4).

For the BAMM analysis, we started estimating empirical priors using the setBAMMpriors function from BAMMtools v2.1.9, and launched BAMM on a reduced phylogeny for the 416 species in our dataset with species-specific sampling fractions determined at the genus level and default parameters for other arguments. We then performed a STRAPP analysis[71] test with the BAMMtools function traitDependentBAMM, to estimate if rates differ between species using and not using trophallaxis. We used the resulting $P$ value to assess whether significant evidence of a trait-dependent effect was detected and the mean diversification rates across species engaging or not with trophallaxis to quantify the magnitude of that difference.

### Statistics and reproducibility

Statistical analyses are described in the corresponding sections of the methods. Source data, trees and code to reproduce the analyses are available in the Zenodo repository[90] and further details can be obtained from the corresponding author. Parameters are provided in the code corresponding to each analysis.

### Reporting summary

Further information on research design is available in the Nature Portfolio Reporting Summary linked to this article.

### Data availability

We provide our datasets, trees and code to conduct these analyses in the GitHub repository https://github.com/Social-Fluids-Lab/Macroevolution_ of_trophallaxis2025 and in the Zenodo repository (https://doi.org/10.5281/ zenodo.14787967)[90]. Any remaining information can be obtained from the corresponding author upon reasonable request.

### Code availability

We provide our datasets, trees and code to conduct these analyses in the GitHub repository https://github.com/Social-Fluids-Lab/Macroevolution_

of_trophallaxis2025 and in the Zenodo repository (https://doi.org/10.5281/ zenodo.14787967)[90]. Software versions are systematically provided in the corresponding method sections.

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

## Acknowledgements

We thank Corrie Moreau and Robert Waterhouse for their precious insights on this project, Giorgio Bianchini for his support in using sMap, and Torsten Hauffe for his support on the speciation/extinction analyses. We also thank the expert myrmecologists Claude Lebas, Naoto Idogawa, Aiki Yamada, Peter Slingsby, Riou Mizuno, Juergen Heinze, Alain Lenoir, Volker Nehring, Fabrice Savarit, and Xim Cerda, who kindly took the time to share their knowledge about trophallaxis with us. Finally, we thank Emile Mermillod and Arthur Matte for help with illustrations, and François Brassard for trophallaxis photographs. A.C.L. and M.-P.M. were supported by a Swiss Science Foundation grant (PR00P3_179776) to A.C.L. M.-P.M. was also supported by the University of Fribourg. D.S. acknowledges ETH Zurich for funding, and D.S. received funding from the Swiss National Science Foundation (PCEFP3_187012), and the Swedish Foundation for Strategic Environmental Research MISTRA within the framework of the research program BIOPATH (F 2022/1448).

## Author contributions

M.-P.M., D.S., and A.C.L. conceived the experiments. M.-P.M. and A.C.L. collected the traits data from the literature. M.-P.M. conducted the analyses under the supervision of D.S. and A.C.L. M.-P.M., D.S., and A.C.L. made the figures and wrote the manuscript with comments from all authors.

## Competing interests

The authors declare no competing interests.
