## [Peer Review file · Communications Biology]

Ecological change and conflict reduction led to a social circulatory system in ants

Corresponding Author: Professor Adria LeBoeuf

This manuscript has been previously reviewed at another journal. This document only contains information relating to versions considered at Communications Biology.

Version 0:

Reviewer comments:

Reviewer #1

(Remarks to the Author)

I read this paper with great interest, and I believe it changes how we understand the evolution of superorganisms. The study establishes a robust connection between the evolution of trophallaxis in ants (a “social circulatory system”) and the reduction of conflict and a change to a liquid–sugar diet. The methods were well conceived and included advancements in the use of machine learning to expand the known dataset on trophallaxis in ants.

This study will make a significant contribution to the field of social evolution, and adds a layer to the already tight connection between the co-diversification of angiosperms and ants. I offer some comments below.

Title – The use of “ecological change” to stand in for a change in diet seems like a bit of a stretch. It would help if there were more analyses that dated the evolution of trophallaxis to the diversification of angiosperms and subsequent adoption of a liquid-sugar diet.

Line 45-47 – Prior to reading the current study, my understanding was that gamergates, and worker reproductive autonomy, was clearly a derived condition in ants. This is the argument presented in Peeters and Ito (2001), which suggests that gamergates evolved as a mechanism to extend the life of the colony after the death of the queen. Peeters and Schmidt (2010) also conducted an ancestral trait reconstruction for the ponerine clade and concluded that the most recent common ancestor lacked gamergates (this was from a conference proceedings available online, but it was not peer reviewed). Given how the current study comes to the opposite conclusion, it would help to explain how/why this departs from the previous understanding (this happens later in the discussion, but it would help to set it up stronger earlier in the paper).

Peeters, C., & Ito, F. (2001). Colony dispersal and the evolution of queen morphology in social Hymenoptera. *Annual review of entomology*, 46(1), 601-630.

Lines 105-106 – The sting is rarely used in intracolony conflict. Gamergate species have evolved a range of ritualized dominance behaviors that reduce the chance of death during competitions. Instead, gamergate species are largely predatory, which requires a sting. Most (or all?) ant species with reproductive workers evolved from species where the sting was already present.

Lines 121-123 – I have not heard this argument before. Is there an explanation or something that can be cited to explain how thin abdominal cuticle would not allow ants to have a stinger? Many myrmecine queens have a functional stinger and become physogastric when reproductively active (e.g., *Solenopsis invicta*), which suggests that having thin cuticle does not preclude the use of a stinger.

Line 127 – Here and elsewhere, I had a hard time distinguishing whether the analyses focused on fully mated workers and workers that lay male eggs. It would help to make it clearer when the results refer only to sexual reproduction by mated workers.

Line 146 – Given the lack of experimental evidence, I suggest softening this language (e.g., “These results suggest a causal link between...”)

Line 150 – “Strengthen” should be “strengthened”

286 – *Brachyponera chinensis* is spelled incorrectly

326 – “The” should be “they”

447 – Reverse the order of “it is”

449 – I wonder whether larval trophallaxis could have a stronger role in the evolution of trophallaxis among adults. How would sugary liquids be fed to larvae without trophallaxis? It seems difficult. Some ants get around this by feeding larvae trophic eggs after sugars have presumably been metabolized by adults (e.g., *Aphaenogaster* spp.). I am wondering if you have evidence of any species that perform trophallaxis regularly with larvae but do not engage in trophallaxis with adults?

Reviewer #2

(Remarks to the Author)

This paper explores the impact that a key behavioral innovation, mouth food transfer, has on the ecology and evolution of the ant family. This behavior does not simply involve the transfer of food but also the transfer of other chemicals which result in it forming a key part of information transfer and social regulation for and colonies. In this paper the authors use a systematic approach to the literature to identify where this behavior occurs among the vast diversity of ants. They then test clear hypotheses around the role of this innovative behavior in shaping the evolutionary trajectories of ant clades.

This paper is very well written. It is engaging and easy to read despite the complex nature of the content within. The analyses are exceptionally thorough, and this gives confidence in the interpretation that the authors provide. As a whole, the paper is thought provoking and of wider relevance in the context of transitions to multicellularity and social organization across scales more generally.

I have some minor suggestions for improvements to the manuscript.

Methods

Generally, these were clear, but I would like a little bit more information on how the manual literature search was carried out. The data collection methods do not provide information about which databases were used, what languages were included, and whether an incognito browser was used to avoid search engines modifying returns based on location and search history.

Also, this section provides comment on the taxonomic distribution of the included species, as does Fig S1, but does not comment on the geographic distribution of the final species set, relative to the distribution of ant species; geographic distributions are often highly biased in English language searches.

“Colony size data were retrieved manually from various sources.” What types of sources? Please provide a bit more information, as ‘various sources’ is almost meaningless.

Results

Lines 80-84 mention 7 traits which the authors indicate there is reason to believe may be related to trophallaxis in some way - but they give no citations or rationale for the inclusion of these 7 traits. I realise that the relationships are complex to explain in few words, and are addressed later in detail in the section on causality modelling, but I think it would help the reader to at least include some citations to evidence that these are genuinely proposed interactions, and not just de novo hypotheses which could be straw men set up for the paper.

Discussion.

The discussion of the papers findings is clear and well-written. The comparison with wasps is interesting, but I would have expected to see a little comparison with behaviours in other groups too, such as bees who also pass food mouth to mouth within the colony, and termites, who make an interesting comparator, given that they are behaviourally in some ways similar to ants, but have a different evolutionary origin.

I think the final sentence would work better without the final 7 words, which suddenly bring in humans without integration into the main body of the text.

Reviewer #3

(Remarks to the Author)

This manuscript aims to reconstruct the evolutionary history of mouth-to-mouth trophic exchange behavior (stomodeal trophallaxis) and other traits such as colony size in ants with reference to their molecular phylogeny. The authors also aim at testing hypotheses on the causal relationship between environmental factors and stomodeal trophallaxis by phylogenetic comparative analyses. They also used AI to complement missing data. These are ambitious endeavors. In the estimation of

the evolutionary timing of stomodeal trophallaxis it was inferred to have occurred twice in the past, but is frequently missed in various taxa. These conclusions seem to be sound based on the current data. Importantly, both the utilization of sugary diets linked to the timing of flowering plant evolution and the decline of workers' reproductive potential were shown to be positively correlated with the evolution of stomodeal trophallaxis. The negative relationship between worker reproductive capacity and stomodeal trophallaxis, the authors believed, supports the hypothesis that the reduction in potential intracolony conflicts was responsible for the evolutionary transitions to superorganism and its function, nutrient circulation. Causation inferences also suggested that trophallaxis also contributed to ant diversification and increased colony size.

As the authors state, it is challenging to study the macroevolutionary history of behavior and its consequences on evolution. The aim of this study is positively evaluated, because the evolution of ant trophallaxis may have contributed to structural changes in terrestrial ecosystems. However, I have a major concern in this study. Surprisingly, there is no mention of a key trait that is widely and commonly found in ants, i.e. nutritional exchange behavior other than stomodeal trophallaxis. Besides the recently discovered pupal-secreted "ant milk" cited in the manuscript, there are other well-known nutrient exchanges: trophic egg, larval hemolymph feeding (LHF) as known in Amblyoponinae, Leptanillinae and Ectatomminae and brood cannibalism known in many ants. Whichever method is used, the trophic cycling, the bloodstream of the colony as a superorganism, can function. The authors' interpretation of the relationship between the evolution of superorganismality and the evolution of stomodeal trophallaxis is therefore questionable.

The following alternative hypotheses can be set out regarding the relationship between trophic egg-laying by workers and mouth-to-mouth nutrient exchange behavior. Ants may always use one of the above-mentioned modes of nutrient exchanges. Perhaps brood cannibalism, trophic eggs and LHF, might be more ancestral than stomodeal trophallaxis. As some ancestral ants using trophic eggs became dependent on nectar along with the evolution of the flowering plants, and as the social stomach and stomodeal trophallaxis evolved, the need for trophic eggs as a means of nutrient cycling within the colony declined and the costly ovarian function of the worker was no longer maintained. According to this scenario, worker ovarian function declined because stomodeal trophallaxis evolved, not because trophallaxis evolved owing to conflict resolution.

In addition, if there is a trade-off between different modes of nutrient exchange, the following hypothesis can be made. In the clades in which stomodeal trophallaxis has been secondarily lost, one of the other ways of nutrition exchanges, such as trophic eggs and LHF may have developed in such clades. In fact, *Pogonomyrmex*, for example, is dependent on trophic eggs.

In conclusion, it would be desirable to reanalyze the data taking into account information on nutritional exchange behaviors other than stomodeal trophallaxis. I know that the empirical data on the other types of nutritional exchanges, such as trophic eggs, LHF, and cannibalism are much sparser than those on stomodeal trophallaxis behavior. Even if the data are too thin to analyze, it would mislead by totally ignoring trophic eggs and LHF at all, as the current manuscript does. This would make the authors' narrative less convincing.

Kazuki Tsuji

Version 1:

Reviewer comments:

Reviewer #1

(Remarks to the Author)

I found the response to my review and others to be thorough and compelling. I am excited about the contribution this paper will make to our field.

Reviewers' comments:

Reviewer #1 (Remarks to the Author):

I read this paper with great interest, and I believe it changes how we understand the evolution of superorganisms. The study establishes a robust connection between the evolution of trophallaxis in ants (a “social circulatory system”) and the reduction of conflict and a change to a liquid–sugar diet. The methods were well conceived and included advancements in the use of machine learning to expand the known dataset on trophallaxis in ants.

This study will make a significant contribution to the field of social evolution, and adds a layer to the already tight connection between the co-diversification of angiosperms and ants. I offer some comments below.

We thank Reviewer 1 for their comments and appreciation. These were also the features we are most excited about!

Title – The use of “ecological change” to stand in for a change in diet seems like a bit of a stretch. It would help if there were more analyses that dated the evolution of trophallaxis to the diversification of angiosperms and subsequent adoption of a liquid-sugar diet.

Dating the two phylogenies is a highly challenging task due to the size of the clades and the relative sparsity of their fossil records and because there are ongoing debates about the time of origin of major clades like flowering plants (Benton, Michael J., Peter Wilf, and Hervé Sauquet. "The Angiosperm Terrestrial Revolution and the origins of modern biodiversity." *New Phytologist* 233.5 (2022): 2017-2035; Silvestro D, Bacon CD, Ding W, Zhang Q, Donoghue PC, Antonelli A, Xing Y. Fossil data support a pre-Cretaceous origin of flowering plants. *Nature ecology & evolution*. 2021 Apr;5(4):449-57). Yet, based on the available data we found that indeed the first transitions to trophallaxis correspond with phases of angiosperm expansion based on recent molecular clock dating (Ramírez-Barahona et al 2020) and on fossil-based estimates that infer a peak in angiosperm family-level diversification rates in the early Cretaceous, around 130 Ma (Silvestro et al. 2021 NEE). This indeed corresponds temporally with the first major gain of trophallaxis in ants as inferred in our study.

We have added additional citations, in particular, Ramírez-Barahona, S., Sauquet, H. & Magallón, S. The delayed and geographically heterogeneous diversification of flowering plant families. *Nat Ecol Evol* 4, 1232–1238 (2020), which describes two stages of angiosperm diversification that correspond well to our findings.

We have added (L249-255):

Dating these two phylogenies is a highly challenging task due to the size of the clades, the relative sparsity of their fossil records, and given that the time of origin of flowering plants remains hotly debated (55,56). Yet, based on available data, the first transition to trophallaxis corresponds with phases of angiosperm expansion based on recent molecular clock dating (56) and on fossil-based estimates that infer a peak in angiosperm family-level diversification rates in the early Cretaceous, around 130 Ma (55).

Line 45-47 – Prior to reading the current study, my understanding was that gamergates, and worker reproductive autonomy, was clearly a derived condition in ants. This is the argument presented in Peeters and Ito (2001), which suggests that gamergates evolved as a mechanism to extend the life of the colony after the death of the queen. Peeters and Schmidt (2010) also conducted an ancestral trait reconstruction for the ponerine clade and concluded that the most recent common ancestor lacked gamergates (this was from a conference proceedings available online, but it was not peer reviewed). Given how the current study comes to the opposite conclusion, it would help to explain how/why this departs from the previous understanding (this happens later in the discussion, but it would help to set it up stronger earlier in the paper).

Peeters, C., & Ito, F. (2001). Colony dispersal and the evolution of queen morphology in social Hymenoptera. *Annual review of entomology*, 46(1), 601-630.

We thank the reviewer for bringing this up and for suggesting that we point this out earlier and more explicitly. Our perspective is indeed somewhat against the current dogma, but as we now describe in an elaborated paragraph in the introduction, there is good reason to reconsider this view (L36-54).

Behavioral traits and associated phenotypic adaptations were central to the evolution of one of the most ecologically dominant invertebrate clades: the ants^{3,8,9}. The ant clade harbors highly diverse ecologies, morphologies, life-history traits, and behaviors. Some ant species display such high levels of cooperation that they are considered to have passed a major evolutionary transition to superorganismality, characterized by firm reproductive division of labor, low conflict and high cooperation¹⁰⁻¹². However, other extant ant species have significant within-colony conflict, where workers can be mated, lay diploid eggs that can develop into new queens, and fundamentally act as the colony's reproductive¹³. Such workers display what we term worker reproductive autonomy, a condition that would threaten colony cohesion, and they do not satisfy Wheeler's conditions for a superorganism¹⁴ (worker lifetime unmatedness). The current dominant view in ant evolution is that reproductive division of labor was already complete in the common ancestor to all ants¹⁵⁻¹⁷ and any species with totipotent workers are secondary reductions. However, the analyses these conclusions were based on relied on strong unvalidated assumptions and at a time when the ant phylogeny was less resolved (a major subfamily with worker reproductive autonomy, Ectatomminae, was misplaced). Further, once systems undergo major evolutionary transitions in individuality¹⁸, various ratcheting processes typically block them from reverting¹⁹. Together, these features collectively suggest we should consider the possibility that not all ants are superorganisms, and instead superorganismality may have evolved one or more times within the ants.

One of us (ACL) is currently working on a longer-form argument in favor of this view.

Lines 105-106 – The sting is rarely used in intracolony conflict. Gamergate species have evolved a range of ritualized dominance behaviors that reduce the chance of death during competitions. Instead, gamergate species are largely predatory, which requires a sting. Most (or all?) ant species with reproductive workers evolved from species where the sting was already present.

We thank the reviewer for pointing this out. While it is rare, the sting is used in dominance battles. Here are two papers with quotes mentioning this.

Description of sting smearing (sting based threats) in *Dinoponera quadricaps* in the context of overthrowing the reproductive gamergate.

Thibaud Monnin, Christian Peeters, Dominance hierarchy and reproductive conflicts among subordinates in a monogynous queenless ant, Behavioral Ecology, Volume 10, Issue 3, May 1999, Pages 323–332, <https://doi.org/10.1093/beheco/10.3.323>

“The active worker curled her gaster forward and extruded her sting, positioning it close to the target and often rubbing it on the cuticle of the target, but never stinging her.

Description of dominance battles amongst queens of *Leptothorax*: Heinze, J., Lipski, N., & Hölldobler, B. (1992). Reproductive competition in colonies of the ant *Leptothorax gredleri*. Ethology, 90(4), 265-278

“However, between 2% and 20% of all contests escalated into heavy mandible and sting fights lasting for 2 min and longer, during which one or both queens stridulated vehemently.

We have adjusted the sentence that the reviewer mentioned to clarify that this is only one of many such behaviors, but may nonetheless be linked with reproductive conflict (though as the reviewer suggests, this link is not supported by the data) L114-116:

“Additionally, because the sting can be used, among other ritualized behaviors, during dominance battles in species with worker reproductive autonomy (34–37), it is expected to indirectly negatively correlate with trophallaxis.”

Lines 121-123 – I have not heard this argument before. Is there an explanation or something that can be cited to explain how thin abdominal cuticle would not allow ants to have a stinger? Many myrmecine queens have a functional stinger and become physogastric when reproductively active (e.g., *Solenopsis invicta*), which suggests that having thin cuticle does not preclude the use of a stinger.

We appreciate the reviewer's point. Broadly, ants with a predominantly sugar diet have thinner cuticles (we are aware that Matte and Economo have a manuscript near submission on this topic). Higher cuticle investment has been associated with a nitrogen rich-diet, either through predation or endosymbionts (Hu, Y., Sanders, J.G., Łukasik, P. et al. Herbivorous turtle ants obtain essential nutrients from a conserved nitrogen-recycling gut microbiome. *Nat Commun* 9, 964 (2018), and has been shown to have a phylogenetic component, with thinner cuticles in the formicoids (Christian Peeters, Mathieu Molet, Chung-Chi Lin, Johan Billen, Evolution of cheaper workers in ants: a comparative study of exoskeleton thickness, *Biological Journal of the Linnean Society*, Volume 121, Issue 3, July 2017, Pages 556–563).

We have adjusted this sentence to clarify this point and include these citations (L130-134):

Finally, in scenario 4 (morphological constraints, Figure 1D) the sting may not be supportable when ants shift to a low-nitrogen sugar diet and can produce only a thin cuticle^{30,45}. In this scenario, reduced intracolony conflict led to reduced need for a sting, which then allowed these lineages to take up a sugary diet and acquire trophallaxis.

Line 127 – Here and elsewhere, I had a hard time distinguishing whether the analyses focused on fully mated workers and workers that lay male eggs. It would help to make it clearer when the results refer only to sexual reproduction by mated workers.

Thank you, yes, we have tried to clarify this throughout at several points in the text.

Line 146 – Given the lack of experimental evidence, I suggest softening this language (e.g., “These results suggest a causal link between...”)

done.

Line 150 – “Strengthen” should be “strengthened”

done

286 – *Brachyponera chinensis* is spelled incorrectly

done

326 – “The” should be “they”

done

447 – Reverse the order of “it is”

done

449 – I wonder whether larval trophallaxis could have a stronger role in the evolution of trophallaxis among adults. How would sugary liquids be fed to larvae without trophallaxis? It seems difficult. Some ants get around this by feeding larvae trophic eggs after sugars have presumably been metabolized by adults (e.g., *Aphaenogaster* spp.). I am wondering if you have evidence of any species that perform trophallaxis regularly with larvae but do not engage in trophallaxis with adults?

We would have loved to also analyze adult-larval trophallaxis, but unfortunately the literature is too sparse in the documentation of this behaviour to do anything meaningful. We have another paper in the pipeline (Matte & LeBoeuf

biorxiv) that builds proxies for adult-larva trophallaxis using larval morphology, so we hope that we can address this in the future. For now, we mention the data limitation in the first paragraph of the results on L98

While adult-larval trophallaxis¹⁴, cuticle thickness³⁵, use of trophic eggs^{36,37}, and frequency of polygyny³⁸ might also be related traits, too few data were available to perform meaningful tests over a clade of >14,000 species.

And in the discussion after mentioning the proposed scenarios for the evolution of trophallaxis in wasps, bees and termites (following Review 2's suggestion) L467-472:

Given that trophallaxis evolved in each of these groups in connection to parental care, it highlights the unusual case of the ants. In ants, because the data on larval-adult trophallaxis are so sparse²⁰, we were not able to explore the possibility that another social transfer may have been the precursor in the evolution of stomodeal trophallaxis between adults. That said, most predatory ants do not feed larvae in individualized interactions like trophallaxis⁸⁴, and we and others clearly find that the common ancestor to all ants was predatory^{32,85}.

Regarding trophic eggs, please see our response to reviewer 3 and the new paragraph in the discussion.

Reviewer #2 (Remarks to the Author):

This paper explores the impact that a key behavioral innovation, mouth food transfer, has on the ecology and evolution of the ant family. This behavior does not simply involve the transfer of food but also the transfer of other chemicals which result in it forming a key part of information transfer and social regulation for and colonies. In this paper the authors use a systematic approach to the literature to identify where this behavior occurs among the vast diversity of ants. they then test clear hypotheses around the role role of this innovative behavior in shaping the evolutionary trajectories of ant clades.

This paper is very well written. It is engaging and easy to read despite the complex nature of the content within. The analyses are exceptionally thorough, and this gives confidence in the interpretation that the author's provide. As a whole, the paper is thought provoking and of wider relevance in the context of transitions to multicellularity and social organization across scales more generally.

We thank reviewer 2 for their comments and appreciation! We are glad that our goals were met.

I have some minor suggestions for improvements to the manuscript.

Methods

Generally, these were clear, but I would like a little bit more information on how the manual literature search was carried out. The data collection methods do not provide information about which databases we used, what languages were included, and whether an incognito browser was used to avoid search engines modifying returns based on location and search history.

Also, this section provides comment on the taxonomic distribution of the included species, as does Fig S1, but does not comment on the geographic distribution of the final species set, relative to the distribution of ant species; geographic distributions are often highly biased in English language searches.

"Colony size data were retrieved manually from various sources." What types of sources? Please provide a bit more information, as 'various sources' is almost meaningless.

We have added a sentence to the methods to describe how we conducted the search for trophallaxis (L521-524). Unfortunately when we began this project we did not use incognito mode for our searches.

We searched Google Scholar for combinations of genus and/or species with “trophallaxis”, “oecotrophobiosis”, “regurgitation”, “share liquid”, which are means by which trophallaxis has been described in scientific literature.

Regarding geographic distribution, given that we have not focused on the relationship between geography and trophallaxis, we think refocusing our dataset to balance across the geographic distribution of ants is beyond the scope of the paper. While this may be a limitation, we do not think it should impact our findings.

Regarding the comment on colony size, yes, absolutely. We have adjusted that methods section (L575-578):

Colony size data were retrieved manually from scientific literature, AntWiki.org, and online shops specialized in ants when not found in other sources. We searched Google Scholar and Google using a combination of the genus and/or species with “colony size” and “colony” as keywords.

Results

Lines 80-84 mention 7 traits which the authors indicate there is reason to believe may be related to trophallaxis in some way - but they give no citations or rationale for the inclusion of these 7 traits. I realise that the relationships are complex to explain in few words, and are addressed later in detail in the section on causality modelling, but I think it would help the reader to at least include some citations to evidence that these are genuinely proposed interactions, and not just de novo hypotheses which could be straw men set up for the paper.

Yes, thank you for this suggestion. We have added a series of citations addressing these points on L98.

Discussion.

The discussion of the papers findings is clear and well-written. The comparison with wasps is interesting, but I would have expected to see a little comparison with behaviours in other groups too, such as bees who also pass food mouth to mouth within the colony, and termites, who make an interesting comparator, given that they are behaviourally in some ways similar to ants, but have a different evolutionary origin.

Yes, we thank the reviewer for this suggestion. We have now added some additional sentences to this effect (L458-466):

Our scenario for the evolution of social regurgitation between adults in ants differs from the hypothesized evolutionary paths for trophallaxis in wasps, bees, and termites^{29,77-79}. In wasps, trophallaxis likely evolved from adults pre-chewing prey and providing it to larval young, and generally, adult-adult trophallaxis only occurs in highly eusocial wasps⁷⁷. In bees, trophallaxis also likely first arose in the context of nutrient provisioning for young⁷⁸, but has also been linked to reproductive conflict. Indeed, in several bee species asymmetrical trophallactic relationships are key to reproductive dominance hierarchies^{23,29}. In termites, parent-to-offspring proctodeal trophallaxis was a key evolutionary innovation for their symbiont-enabled lifestyle^{4,79}. None of these systems have been evaluated in the phylogenetic detail we have done here.

I think the final sentence would work better without the final 7 words, which suddenly bring in humans without integration into the main body of the text.

Yes, done.

Reviewer #3 (Remarks to the Author):

This manuscript aims to reconstruct the evolutionary history of mouth-to-mouth trophic exchange behavior (stomodeal trophallaxis) and other traits such as colony size in ants with reference to their molecular phylogeny. The authors also aim at testing hypotheses on the causal relationship between environmental factors and stomodeal trophallaxis by phylogenetic comparative analyses. They also used AI to complement missing data. These are ambitious endeavors. In the estimation of the evolutionary timing of stomodeal trophallaxis it was inferred to have occurred twice in the past, but is frequently missed in various taxa. These conclusions seem to be sound based on the current data. Importantly, both the utilization of sugary diets linked to the timing of flowering plant evolution and the decline of workers' reproductive potential were shown to be positively correlated with the evolution of stomodeal trophallaxis. The negative relationship between worker reproductive capacity and stomodeal trophallaxis, the authors believed, supports the hypothesis that the reduction in potential intracolony conflicts was responsible for the evolutionary transitions to superorganism and its function, nutrient circulation. Causation inferences also suggested that trophallaxis also contributed to ant diversification and increased colony size.

As the authors state, it is challenging to study the macroevolutionary history of behavior and its consequences on evolution. The aim of this study is positively evaluated, because the evolution of ant trophallaxis may have contributed to structural changes in terrestrial ecosystems. However, I have a major concern in this study. Surprisingly, there is no mention of a key trait that is widely and commonly found in ants, i.e. nutritional exchange behavior other than stomodeal trophallaxis. Besides the recently discovered pupal-secreted "ant milk" cited in the manuscript, there are other well-known nutrient exchanges: trophic egg, larval hemolymph feeding (LHF) as known in Amblyoponinae, Leptanillinae and Ectatomminae and brood cannibalism known in many ants. Whichever method is used, the trophic cycling, the bloodstream of the colony as a superorganism, can function. The authors' interpretation of the relationship between the evolution of superorganismality and the evolution of stomodeal trophallaxis is therefore questionable.

The following alternative hypotheses can be set out regarding the relationship between trophic egg-laying by workers and mouth-to-mouth nutrient exchange behavior. Ants may always use one of the above-mentioned modes of nutrient exchanges. Perhaps brood cannibalism, trophic eggs and LHF, might be more ancestral than stomodeal trophallaxis. As some ancestral ants using trophic eggs became dependent on nectar along with the evolution of the flowering plants, and as the social stomach and stomodeal trophallaxis evolved, the need for trophic eggs as a means of nutrient cycling within the colony declined and the costly ovarian function of the worker was no longer maintained. According to this scenario, worker ovarian function declined because stomodeal trophallaxis evolved, not because trophallaxis evolved owing to conflict resolution.

In addition, if there is a trade-off between different modes of nutrient exchange, the following hypothesis can be made. In the clades in which stomodeal trophallaxis has been secondarily lost, one of the other ways of nutrition exchanges, such as trophic eggs and LHF may have developed in such clades. In fact, *Pogonomyrmex*, for example, is dependent on trophic eggs.

In conclusion, it would be desirable to reanalyze the data taking into account information on nutritional exchange behaviors other than stomodeal trophallaxis. I know that the empirical data on the other types of nutritional exchanges, such as trophic eggs, LHF, and cannibalism are much sparser than those on stomodeal trophallaxis behavior. Even if the data are too thin to analyze, it would mislead by totally ignoring trophic eggs and LHF at all, as the current manuscript does. This would make the authors' narrative less convincing.

Kazuki Tsuji

Thank you, Prof. Tsuji.

This is very insightful and we are grateful that you've brought up these topics, giving us space and context to mention these ideas and amazing features of ant biology. As we have written (Negroni & LeBoeuf 2023), all of the mentioned behaviours can indeed underlie metabolic division of labor within colonies. However, the reliance on metabolic

division of labor does not necessarily lead to a major evolutionary transition in the level of the individual. For example, through lactation, mammalian mothers support offspring through metabolic division of labor, yet we do not create a new level of individual. Whether these other social transfer behaviors (LHF and trophic eggs) can underlie an integrated organismal level are fascinating questions worthy of future study. At the moment the conceptual framework to parse these bits of biology has scarcely been formed (Hakala et al. 2023). For example, when does worker policing transform into trophic egg production? In a society of individuals, the laying of eggs is likely for selfish opportunistic interests that may then be exploited by others, and may be disguised in dishonest signals. If egg laying were to evolve in workers to underlie a social circulatory system after the transition to superorganismality, one would expect them not to be viable and to have more variable contents and honest signals. One of the ways we have been conceptualizing this shift (Hakala et al. 2023) is to go by the definition of a socially transferred material – when material is metabolized by the donor, transferred to another individual where it acts, and through this action, **benefits the donor**. In the evolution of trophic eggs, these questions are very interesting, very open and very understudied. Adult-adult trophallaxis is one of the few behaviors in the suite we've been discussing where donors are clearly willing (as opposed to the non-destruction cannibalism in LHF), making the evolutionary context simpler to interpret. In summary, these social transfer behaviors (trophallaxis, LHF, trophic eggs) have a lot in common, but to adequately address LHF and trophic eggs would require more clarity on the role these social transfers play within colonies across species and we believe that to do this well would be more far reaching than this paper can cover.

Regarding further analysis on LHF and trophic eggs: As you have mentioned, the data are indeed sparse. Based on Genzoni et al. 2023 and papers we are aware of on LHF, we have had a look at the coincidence between trophic eggs and trophallaxis and LHF and trophallaxis, extrapolating species data to genus level (Supplementary figure S8, next page). We have now mentioned trophic eggs when describing the things we would've liked to have analysed given more data L98 and we have also added the paragraph below to the discussion (L477-491):

Alternative modes of nutrient exchange such as feeding on larval hemolymph through a larval hemolymph tap, tubercle or puncture, or producing trophic eggs, could serve a similar purpose as trophallaxis in creating a network of metabolic division of labor across the colony^{20,36}. However, it remains difficult to test this hypothesis due to the scarcity of data, especially in the annotation of species that do not use these modes of nutrient transfer. When these sparsely annotated traits are viewed in the context of trophallaxis evolution (Supplementary Figure S8, from data in Genzoni et al. 2023³⁷ and our social exchanges database⁸⁶ extrapolated to genus level), there is no clear interaction between use of trophic eggs and trophallaxis, but larval hemolymph feeding is only found to occur in species not using trophallaxis. This could mean that larval hemolymph feeding might be an ancestral nutrient exchange behavior and/or a behavior secondarily evolved in genera that lost trophallaxis. While we found no clear correlation between trophic eggs and trophallaxis, this could be due to true biology or insufficiently precise trait annotation. Indeed, there are several species of Dolichoderinae and some Myrmicinae that are highly reliant on trophic eggs that would phylogenetically be expected to perform trophallaxis but do not^{87,88}. This is a fascinating domain for future study.

Indeed, we wanted to test for phylogenetic correlation across the traits, but given that we have next to no clear “does not use” trait annotation, any analysis we do is unlikely to be meaningful (perhaps all ants use trophic eggs; a model cannot distinguish missing data from ‘does not use’). This was a major challenge in our analysis of trophallaxis and likely will be for future comparative phylogenetic analysis of trophic egg use.

In summary, we hope that the additional figure and paragraph in the discussion are sufficient to highlight that this topic is worthy of further exploration but we do not have the capacity to address it adequately with available knowledge.

Figure S8 Records of genera in which adults drink larval hemolymph and/or produce trophic eggs. The tree represents the ancestral state reconstruction of trophallaxis on the MCC tree for 417 ant species. The branch color reflects the posterior probability of trophallaxis. We highlight whether the trophallaxis behavior of extant species is imputed (light grey). We extrapolated from species records to the genus level whether adults drink larval hemolymph and/or produce trophic eggs. In green, at least one species has the trait, in maroon, at least one species does not, and in red at least one species does not have the trait.